# Genome-Wide Analysis of the Odorant Receptor Gene Family in *Solenopsis invicta*, *Ooceraea biroi*, and *Monomorium pharaonis* (Hymenoptera: Formicidae)

**DOI:** 10.3390/ijms24076624

**Published:** 2023-04-01

**Authors:** Bo Zhang, Rong-Rong Yang, Xing-Chuan Jiang, Xiao-Xia Xu, Bing Wang, Gui-Rong Wang

**Affiliations:** 1State Key Laboratory for Biology of Plant Diseases and Insect Pests, Institute of Plant Protection, Chinese Academy of Agricultural Sciences, Beijing 100193, China; 2Shenzhen Branch, Guangdong Laboratory for Lingnan Modern Agriculture, Genome Analysis Laboratory of the Ministry of Agriculture, Agricultural Genomics Institute at Shenzhen, Chinese Academy of Agricultural Sciences, Shenzhen 518120, China; 3College of Plant Protection, Anhui Agricultural University, Hefei 230036, China; 4Laboratory of Bio-Pesticide Creation and Application of Guangdong Province, College of Plant Protection, South China Agricultural University, Guangzhou 510642, China

**Keywords:** *Solenopsis invicta*, *Ooceraea biroi*, *Monomorium pharaonis*, odorant receptor, tandem duplication, selective evolutionary pressure, expression profiles, monogyne, polygyne

## Abstract

Olfactory systems in eusocial insects play a vital role in the discrimination of various chemical cues. Odorant receptors (ORs) are critical for odorant detection, and this family has undergone extensive expansion in ants. In this study, we re-annotated the *OR* genes from the most destructive invasive ant species *Solenopsis invicta* and 2 other Formicidae species, *Ooceraea biroi* and *Monomorium pharaonis*, with the aim of systematically comparing and analyzing the evolution and the functions of the *ORs* in ant species, identifying 356, 298, and 306 potential functional *ORs*, respectively. The evolutionary analysis of these *ORs* showed that ants had undergone chromosomal rearrangements and that tandem duplication may be the main contributor to the expansion of the *OR* gene family in *S. invicta*. Our further analysis revealed that 9-exon *ORs* had biased chromosome localization patterns in all three ant species and that a 9-exon *OR* cluster (*SinvOR4–8*) in *S. invicta* was under strong positive selection (Ka/Ks = 1.32). Moreover, we identified 5 *S. invicta OR* genes, namely *SinvOR89*, *SinvOR102*, *SinvOR352*, *SinvOR327*, and *SinvOR135*, with high sequence similarity (>70%) to the orthologs in *O. biroi* and *M. pharaonis*. An RT-PCR analysis was used to verify the antennal expression levels of these *ORs*, which showed caste-specific expression. The subsequent analysis of the antennal expression profiles of the *ORs* of the *S. invicta* workers from the polygyne and monogyne social forms indicated that *SinvOR35* and *SinvOR252* were expressed at much higher levels in the monogyne workers than in the polygyne workers and that *SinvOR21* was expressed at higher levels in polygyne workers. Our study has contributed to the identification and analysis of the *OR* gene family in ants and expanded the understanding of the evolution and functions of the ORs in Formicidae species.

## 1. Introduction

The formation of new genes, which has contributed tremendously to the evolution of developmental programs in various organisms, is a major reason for the generation of biodiversity [1,2]. New genes can be formed in many ways, including through gene duplication, exon-shuffling, gene fission–fusion, and de novo origination [3]. The genes derived from the same ancestor, called homologous genes, are divided into three types: orthologs, xenologs, and paralogs. Orthologs are genes that have evolved as a result of speciation and still retain functions related to those of the ancestral gene. Xenologs are produced via the transfer between species and separated by a large evolutionary span via symbiosis or viral infection (i.e., horizontal gene transfer) [4]. The primary mechanism of gene family expansion is through gene duplication. These duplicated genes are referred to as paralogs to indicate that their homology related a gene duplication event, rather than a speciation event [5].

Gene families are groups of genes descended from a common ancestor that retain similar sequences and functions [6]. They have obvious similarities in structure and function and encode similar proteins. There are five mechanisms contributing to the formation of gene families: whole genome duplication, tandem duplication, transposon-mediated duplication, segmental duplication, and retro-transposition [7]. Although the processes leading to gene duplication are fairly well understood, it has been difficult to tease apart the role of each process in the evolution of different gene families, especially those that are rapidly evolving [1].

The odorant receptor (*OR*) gene family in Hymenoptera insects is a good model system to study gene evolution, especially in the Formicidae species. The olfactory system plays an important role in allowing insects to sense environmental chemicals. This system guides insects towards food, mating partners, and oviposition locations, and it also helps them detect predators, identify toxic compounds, and communicate socially [8]. Signal transduction in the insect peripheral olfactory system consists of the following steps. First, external odorant molecules enter the lymph through the polar pores of the cuticle and then are bound by surrounding odorant-binding proteins (OBPs). Second, with the help of OBPs, odorant molecules are transported to the dendritic membranes of olfactory sensory neurons (OSNs) and activate ORs or ionotropic receptors expressed on OSNs. Next, the chemical signals are converted into electrical signals and transmitted to the antennal lobe in order to activate the central nervous system, which then directs the corresponding behavioral responses. Finally, the odorant molecules degrade [9,10].

Ants have the largest olfactory gene families reported among all insects with sequenced genomes; for example, there are 320 *ORs* in *Linepithema humile*, 291 *ORs* in *Pogonomyrmex barbatus*, and 461 *ORs* in *Harpegnathos saltator* [11,12]. Several studies have shown that most *ORs* in different ant species were quite young. These *ORs* were created about 150 million years ago (MYA) by a large number of tandem duplication events, following the divergence of ants from their Hymenopteran cousins, the bees [13,14,15]. Moreover, ants have evolved a large *OR* subfamily called the 9-exon *OR* subfamily, which has more genes than any other *OR* subfamily [12,15]. Evolutionary analyses showed that the 9-exon *OR* clade underwent a striking amount of expansion in the ancestors of all ant species and has continued to expand slowly in extant ant lineages [13]. A functional study showed that 9-exon *ORs* could usually sense cuticular hydrocarbons (CHCs) that were derived from the body surface of the insects and were used to exchange chemical information among ants [16]. For example, ants can distinguish individuals from intraspecific and interspecific nests by recognizing the CHCs [17,18]. However, there have been few studies of the mechanism by which diverse CHCs were detected by 9-exon *ORs* [13,19,20].

*S. invicta* is native to South America and concentrated in the Pantanal region, at the headwaters of the Paraguay River [21,22]. This species has recently invaded Australia, China, and Japan [23], and it has become one of the most destructive invasive species in the world, threatening biodiversity and agricultural production. *S. invicta* ants have two types of societies: single-queen (monogyne) and multi-queen (polygyne) societies. Monogyne queens have larger fat reserves and fly higher and farther than polygyne queens during marriage flight, and they are also more likely to invade new areas to build nests, whereas polygyne queens are more likely to join mature colonies [24]. Several studies have shown that the divergence of the social forms is determined by the green-beard gene *general* protein-*9* (*Gp-9*) alleles in both workers and queens [25,26,27]. The *Gp-9* gene encodes a member of the OBP subfamily [28,29]. In monogyne colonies, the queens have a homozygous B-allele at the locus *Gp-9*(BB), while queens in polygyne colonies have the genotype *Gp-9*(Bb). The workers carrying the *Gp-9*(Bb) genotype recognize a specific CHC secreted by a queen with a b-allele at the *Gp-9* locus. With the assistance of *Gp-9*, this unknown compound activates *OR*-expressed OSNs and guides the Bb workers to kill the BB queens, while *Gp-9*(Bb) queens are spared [30]. Hence, the workers bearing the allele *Gp-9*(b) may distinguish odorants from BB and Bb queens [31].

Although genome-sequencing technology has greatly improved, it is difficult to identify all ant *ORs* because the gene family is so large. In this study, we re-annotated the *ORs* in *S. invicta*, and *O. biroi*, and annotated the *ORs* in *M. pharaonis* for the first time using publicly available genome data with the aim of further understanding and comparing the evolution and the functions of the *OR* genes in ants [26]. We constructed a phylogenetic tree of the ORs in these three ants and analyzed their chromosomal localization and evolution. Moreover, we analyzed the antennal expression profiles of the *ORs* in the workers, using *S. invicta* transcriptome data [26]. Our systematic analysis of the *ORs* in Formicidae species will assist in future efforts to functionally characterize the *ORs* in order to elucidate the olfactory mechanisms.

## 2. Results

### 2.1. Annotation of the ORs in the Genomes of Three Formicidae Species

Three Formicidae species were selected for a comparative analysis of the *OR* gene family because they had high-quality genome assemblies. The genome size of *S. invicta* (365 Mb) was larger than that of *O. biroi* (216 Mb) and *M. pharaonis* (314 Mb) (Appendix A, Appendix A). Of the BUSCOs, 99.1% (*n* = 1367) were identified in *S. invicta*; 97.4% of the BUSCOs were complete with a single copy, 1.7% were complete and duplicated, and only 0.3% and 0.6% were fragmented and missing, respectively (Appendix A). In the *O. biroi* and *M. pharaonis* genomes, 98.5% and 98.9% of BUSCO genes were present, respectively. The scaffold N50 values were 26.22 Mb in *S. invicta*, 16.89 Mb in *O. biroi*, and 29.66 Mb in *M. pharaonis*, indicating that all three genomes were well assembled (Appendix A). The high integrity of the assembly and the high accuracy of the annotation were beneficial for the gene family analysis.

We used *A. mellifera* as an outgroup to construct a phylogenetic tree of the three Formicidae species using all the longest transcripts of each species. According to the tree, *S. invicta* and *M. pharaonis*, which belong to the Myrmicinae subfamily, diverged from each other around 31.8–44.1 MYA. *O. biroi*, which belongs to the Dorylinae subfamily, diverged from the Myrmicinae around 74.7–100.8 MYA, indicating that it was distantly related to the other 2 ant species (Figure 1).

We obtained a set of intact *ORs* that were putatively functional by filtering out the pseudogenes. We annotated more intact *OR* genes, as compared to previous studies [32,33]. For example, we annotated 356 intact *ORs* with 496 alternative spliced variants in *S. invicta*, which was significantly more than the 297 intact *ORs* annotated in the first publication of the *S. invicta* genome [32]. As compared to the 503 alternatively spliced *OR* transcripts annotated in the first version of the *O. biroi* genome [33], we identified 1 additional putatively intact *OR*. We classified them as 298 intact *OR* genes with 504 alternative spliced variants, which slightly improved the previous annotation. We annotated 306 intact *ORs* with 356 alternative spliced variants in the *M. pharaonis* genome, which had not previously been annotated.

### 2.2. Genomic Organization of ORs

Almost all the *ORs* could be mapped to chromosomes because of the high quality of the *S. invicta*, *O. biroi*, and *M. pharaonis* genomes. We renamed these *ORs* according to their order on the chromosomes. The *ORs* that could not be mapped to a chromosome (*SinvOR355*, *SinvOR356*, and *MphaOR299*–*MphaOR306*) were named according to the order in which they appeared in the general feature format file. The chromosomal mapping of the *OR* genes from *S. invicta* showed that the 354 *ORs* were located on all chromosomes except for Chromosome 9 (Chr9) (Figure 2a). Similarly, in *O. biroi OR* genes were present on all chromosomes except Chr13 (Figure 2b). However, *OR* genes were present on all chromosomes in *M. pharaonis* (Figure 2c).

We found that most of the *ORs* in the 3 Formicidae species were found in clusters (Figure 2): 335 *ORs* were present in 28 clusters and 19 *ORs* were present as singletons in *S. invicta*; 277 *ORs* were present in 26 clusters and 21 *ORs* were present as singletons in *M. pharaonis*; and 266 *ORs* were present in 34 clusters and 32 *ORs* were present as singletons in *O. biroi*. Next, we counted the number of the exons in all the *ORs* in *S. invicta* to determine whether the *ORs* were derived from transposon-mediated duplication. There was no evidence of recent intron loss, indicating that there had been no transposon-mediated *OR* gene replication. An MCscanX analysis showed that 71 pairs of the *ORs* were tandem duplicates (Appendix A).

### 2.3. The 9-Exon OR Subfamily Expanded in Formicidae Species

Next, we analyzed the expansion of the 9-exon *OR* gene family in the Formicidae species. In total, 89, 118, and 35 9-exon *ORs* were annotated in *S. invicta*, *O. biroi*, and *M. pharaonis*, respectively. There was more expansion of the 9-exon *OR* subfamily in *S. invicta* and *O. biroi* than in *M. pharaonis* (Figure 2). We found that more than 85% of the 9-exon *ORs* in *S. invicta* were present in gene clusters and were found on specific chromosomes, rather than being equally distributed across the genome. A large number of the 9-exon *ORs* were located on Chr1, 5, 6, 7, and 8 of *S. invicta*, whereas none were found on Chr4, 9, 10, 11, or 14 (Figure 2a). In *O. biroi*, the 9-exon *ORs* were located on Chr1, 2, 6, and 10 (Figure 2b).

Next, we calculated the selective evolutionary pressure experienced by the 9-exon *OR* clusters in order to determine the potential reasons for the formation of these clusters, using *S. invicta* as an example. There were 2 tandem clusters of 9-exon *ORs* (*SinvOR4–8*, *SinvOR18–22*) on Chr1. The Ka/Ks value of the cluster *SinvOR4–8* was 1.32, which was greater than 1 (*p* < 0.05, Table 1). This result suggested that this cluster was under strong positive selection. There was no evidence for the positive selection of the other clusters on Chr1, 3, 6, 7, 8, and 16. The Ka/Ks value of the cluster (*SinvOR65–84*) on Chr5 was even higher (3.10) than that of *SinvOR4–8*, but the *p*-value was greater than 0.05, indicating that this could have been a false positive (Table 1).

### 2.4. Genome Synteny Analyses

The analysis of the synteny of the *OR* genes in the three Formicidae species revealed that *S. invicta* and *M. pharaonis* had the most collinear blocks (e.g., a high level of conservation between the first chromosome of *S. invicta* and the second chromosome of *M. pharaonis*), while *S. invicta* and *O. biroi* had the fewest. The analysis of the collinear blocks showed that the chromosomal rearrangements had occurred during the evolution of these species (Figure 3a). We counted the *ORs* in the collinear blocks between these ant species. There were 129 *ORs* in the collinear blocks between *S. invicta* and *M. pharaonis*, and 45 *ORs* in those between *S. invicta* and *O. biroi* (Figure 3a). In total, 36 *ORs* in 3 ant species were collinear. These collinear genes included six 9-exon *ORs* in *S. invicta*, seven 9-exon *ORs* in *O. biroi*, and only one 9-exon *OR* in *M. pharaonis* (Figure 3b).

### 2.5. Phylogenetic Analysis of ORs

A phylogenetic tree of the ORs from the three Formicidae species was constructed. A large clade of the *OR* genes in the phylogenetic tree revealed the expansions of the *OR* gene family, especially in *O. biroi* and *S. invicta* (Figure 4). We found that most of the ORs in this clade were 9-exon ORs, suggesting that the 9-exon *ORs* in ants evolved along with lineage-specific expansions of the *OR* subfamily.

In addition, 17 clades (A–Q) with single-copy orthologs of the ORs in the three species were analyzed (Appendix A). The ORs in 5 of these clades (D, E, G, O, and Q), were significantly conserved, with amino acid similarities greater than 70%. A MEME analysis of the *S. invicta* proteins SinvOR89, SinvOR102, SinvOR352, SinvOR327, and SinvOR135 showed that the motifs in these proteins had been conserved, suggesting they may have important functions (Figure 5).

### 2.6. Analysis of Differentially Expressed OR Genes in S. invicta Workers from Different Societies

Using published data [26], we constructed the antennal expression profiles of the *OR* genes in the worker ants from monogyne SB/SB and polygyne SB/SB colonies (Figure 6, Appendix A). There were differences in the expression levels of the ORs in the antennae of the workers from the monogyne and polygyne *S. invicta* colonies. *SinvOR35* and *SinvOR252* were expressed at much higher levels in the monogyne workers than in the polygyne workers. *SinvOR21* was expressed at higher levels in polygyne workers, but the expression level was still very low.

Three *OR* genes from the conserved clades, *SinvOR89*, *SinvOR102*, and *SinvOR352*, were expressed at a relatively high level in the antennae, while the expression levels of *SinvOR327* and *SinvOR135* were low. The expression levels of all five of these *ORs* in *S. invicta* were analyzed by an RT-PCR analysis of the antennae and legs of five castes of ants: WF, WR, WN, AVQ, and M. All five *ORs* were expressed in the antennae of all five castes, consistent with the data from the transcriptome analysis. *SinvOR102* was expressed at a relatively high level in the antennae of the WR and the WF; *SinvOR89* was expressed at a relatively high level in the antennae of the workers and the AVQ; and *SinvOR327* and *SinvOR352* were highly expressed in the antennae of the WR. The expression levels of *SinvOR135* were consistently low (Figure 7).

At the same time, we also evaluated the 9-exon *ORs* expressed at high levels in the antennae of the workers. We found that *SinvOR78*, *SinvOR120*, and *SinvOR50* were expressed at much higher levels than the other 9-exon *ORs*, and these highly expressed *ORs* may have important functions in distinguishing nest mates and distinguishing between the queen and the workers.

## 3. Discussion

The *OR* gene family in ants has undergone rapid expansion, but little is known about the evolutionary mechanisms of *OR* expansion. A variety of molecular mechanisms lead to gene duplication; the resulting duplicates can be located in tandem clusters or interspersed across the genome. Furthermore, the genomic rearrangement and transposition can disrupt and scatter tandem duplicated genes [5]. Our analysis showed that there was almost no segmental duplication or transposon-mediated duplication during the expansion of the *OR* gene family in *S. invicta*. In addition, no recent intron losses were observed in any ant *OR*, but a large number of tandem repeats were observed, which suggested that the expansion of the *ORs* in *S. invicta* may have occurred via tandem replication. Interestingly, a large number of *OR* tandem repeats have also been found in other Hymenoptera species. For example, *A. mellifera ORs* were found to be mostly located in tandem duplicate arrays, one of which even contained 60 *OR* genes [34]. The discovery of tandem duplicate arrays of the *ORs* in both ants and bees suggests that tandem replication may be the main mechanism of *OR* amplification in the Hymenoptera.

In this study, we observed that some *ORs* in *S. invicta*, *O. biroi*, and *M. pharaonis* were clustered on specific chromosomes, and the chromosomes with these clusters differed between species. However, the *OR* clusters appeared almost exclusively in the Hymenoptera and were not common in other orders [35,36]. Moreover, the *ORs* of three *Drosophila* species, *D. melanogaster*, *D. yakuba*, and *D. pseudoobscura*, were found to be scattered across multiple chromosomes, without obvious clustering [37]. However, mice had a total of 1391 *ORs*, most of which were found within 69 clusters [38]. Hence, the chromosomal distribution of the *ORs* in ants is similar to that in vertebrates; the similar patterns of the genetic evolution involved in olfactory functions (i.e., tandem duplication) may be related to the common eusocial behaviors of ants and vertebrates.

In total, 89, 118, and 35 nine-exon *ORs* were identified in *S. invicta*, *O. biroi*, and *M. pharaonis*, respectively. The 9-exon *ORs* in *S. invicta* and *O. biroi* accounted for approximately one-third of the total *ORs*. The expansion of the 9-exon *OR* subfamily is a common phenomenon in the Hymenoptera. For example, a large number of 9-exon *OR* subfamily members were found in the genome of the army ant *Eciton burchellii,* despite having a reduced gene complement relative to other ants [39]. The 9-exon *OR* subfamily was also large in the paper wasp, accounting for almost half of the Polistes *ORs* [20]. The shaping of the 9-exon *ORs* was partly related to evolutionary pressure [20]. We identified a 9-exon *OR* cluster *SinvOR4–8* that was subject to strong positive selection, which was in accordance with previous studies that had found that the 9-exon *ORs* had experienced slower but continued expansion in modern ant lineages [12,13]. We observed that 9-exon *OR* clusters were located on specific chromosomes in all three ant species, which is a widespread pattern in ant *OR* evolution.

Chemical communication is vital for social insects, particularly via chemical cues, such as a variety of pheromones, mediated foraging, recognition, and reproductive behaviors [40]. For example, trail pheromones are important for foraging behaviors in ant species. Recently, four trail pheromones, *Z*, *E*-α-farnesene (ZEF); *E*, *E*-α-farnesene (EEF); *Z*, *E*-α-homofarnesene (ZEHF); and *E*, *E*-α-homofarnesene (EEHF), were identified in *S. invicta*, and two of them, ZEF and EEF, mediated the mutualism between ants and aphids [41]. In addition, the trail pheromones in Myrmicine and Formicine species were identified as farnesene isomers [42,43], suggesting that the mechanism for recognizing trail pheromones may be similar between species. In this study, we identified 5 *S. invicta ORs*, *SinvOR89*, *SinvOR327*, *SinvOR135*, *SinvOR327*, and *SinvOR135*, that were highly conserved single-copy orthologs in all 3 ant species examined. These ORs may be involved in interspecific recognition of specific chemicals, such as trail pheromones. Further functional studies are needed to test whether these ORs detect farnesene isomers.

The components of the alarm pheromones in ant species have also been identified. For example, 2-ethyl-3,6-dimethyl pyrazine, which had been reported to warn against the predator *Pseudacteon tricuspis*, was isolated from *S. invicta* [44]. A mixture of 4-methyl-3-heptanone and 4-methyl-3-heptanol was identified as an alarm pheromone in *O. biroi* and the close relatives of Eciton army ants [45,46,47]. These results indicated that the components of the alarm pheromones in ants were variable. However, little is known about the molecular basis of the interspecific and intraspecific alarm pheromone detection in ants. The ORs involved in detecting alarm pheromones in *O. biroi* and Eciton army ants may be characterized by searching for conserved clades and comparing the expression levels of the *ORs* in specific tissues of ants from different castes, because ants are more sensitive to alarm pheromones.

Eusocial ants use colony- or species-specific blends of CHCs to convey social information, including nestmate and caste recognition [48] CHCs are detected by OSNs, which are housed in the female-specific basiconic sensilla in the ant antennae [49,50,51], and they are detected by specific members of the expanded 9-exon OR subfamily [48]. Several studies have shown that 9-exon ORs could detect numerous CHCs, including nestmate and non-nestmate CHC blends and queen pheromones, which are necessary for normal nesting behavior and the recognition of nestmates. In general, the *ORs* that function in response to CHCs have been shown to be highly expressed, suggesting that high expression could be used as a criterion to identify these *ORs* in other species. For example, highly expressed *OR HsOR263* in *H. saltator* could respond strongly to the candidate queen pheromone component 13, 23-dimethylheptatriacontane (13,23-DiMeC37), and 15-methylnonacosane (15-MeC29) in cuticular extracts of workers and gamergates, suggesting that this receptor could play an important role in the detection of reproductive individuals [19]. Another study showed that the 9-exon *H. saltator* ORs HsOR271 and HsOR259-L2 were also responsible for the detection of 13,23-DiMeC37 [52]. In addition, non-9-exon ORs were found to detect CHCs with long-chain alkanes. In this study, we identified 3 highly expressed 9-exon *ORs* in *S. invicta*, namely *SinvOR78*, *SinvOR50*, and *SinvOR120*, by constructing *S. invicta* antenna expression profiles of the workers. *SinvOR78* exists in a 9-exon *OR* cluster that is rapidly evolving. This result, combined with previous findings for other ants, suggested that *S. invicta* could sense CHCs, such as nestmate and queen pheromones, through these highly expressed 9-exon *ORs*.

The social types of *S. invicta* were determined by assessing the *Gp-9* genotypes of both the workers and the queens. In this study, we analyzed the expression levels of the *ORs* in the antennae of the *S. invicta* workers from monogyne and polygyne colonies and found that the *ORs SinvOR35*, *SinvOR252*, and *SinvOR21* were differentially expressed among these workers. *SinvOR35* and *SinvOR252* were much more highly expressed in the antennae of the monogyne workers than in the polygyne workers, suggesting that these genes could have key functions in the olfactory detection of workers in the monogyne social type. However, *SinvOR21* was more highly expressed in the polygyne workers, suggesting that this gene could be involved in detecting BB-queen-derived odorants and assist in maintaining the polygyne social type [53].

## 4. Materials and Methods

### 4.1. Insect Rearing and Sample Preparation

Colonies of the red fire ant *S. invicta* were collected from Xintian Village, Jiulong Town, Huangpu District, Guangzhou City, Guangdong Province, China (23°32′34.69″ N, 113°58′27.60″ E). Red ants were reared in the laboratory at 26 ± 2 °C, 70% ± 5% relative humidity, and 12 h light/12 h darkness, and they were fed 10% honey–water and the yellow mealworm *Tenebrio molitor*.

### 4.2. Data Sources

We searched the genomes of all Formicidae species in the NCBI database. Three Formicidae species, namely *M. pharaonis* (GCF_013373865.1; from NCBI), *O. biroi* (GCF_003672135.1; from NCBI), and *S. invicta* (GCF_016802725.1; from NCBI) were selected for further analysis because they had the best chromosome-scale genome assemblies and the most complete gene annotations. Benchmarking Universal Single-Copy Orthologs (BUSCOs, version 5.2.2) was used to assess the quality of the genome assemblies for these three ants; 1367 universal single-copy orthologous genes from insecta_odb10 were used as queries in searches against the assembly using the default parameters [54]. The scaffold N50 length was calculated using Python scripts.

### 4.3. Phylogenetic Analysis

Three Formicidae species, *M. pharaonis*, *O. biroi*, and *S. invicta*, and one outgroup species, *Apis mellifera* (GCF_003254395.2; from NCBI), were selected to construct a phylogenetic tree. The phylogenetic tree was constructed using Orthofinder2 (version 2.5.4, Oxford University, Oxford, United Kingdom) with default parameters [55] and using the longest transcripts for each gene. To estimate species-divergence times, we first queried the divergence time between *A. mellifera* and *S. invicta* on the Time Tree of Life (http://www.timetree.org/, accessed on 10 November 2022) divergence time between 114.9–163.5 MYA) as the calibration point, and then we used MCMCTREE from the PAML package (version 4.8a) to calculate divergence times for each species [56]. The phylogenetic tree was visualized with Evolview (version v2, Northeastern University, Boston, MA, USA) [57].

### 4.4. Analysis of the OR Gene Family

First, all known OR protein sequences of Hymenoptera were downloaded from the NCBI database (https://www.ncbi.nlm.nih.gov/, accessed on 28 March 2022). Pseudogenes were removed manually. Then, the remaining sequences were used as queries in BLASTP searches against the transcripts of *M. pharaonis*, *O. biroi*, and *S. invicta* (BLAST version 2.9.0+, E-value = 1 × 10^−5^) [58]. HMMER (version 3.3.2) searches against the Pfam database were performed to confirm the structural domains of the candidate *OR* genes [59,60]. In brief, for each species, the HMM profile 7tm_6 (PF02949) was used for an hmmsearch against candidate *OR* genes identified by BLAST. Next, the *OR* genes obtained by the preliminary HMMER screen were used to construct separate hidden Markov models for each ant species, and a secondary hmmsearch was conducted to ensure that no *ORs* were missing. Protein sequences identified by the above steps were then used as queries in a BLASTP search against the non-redundant protein sequence database (e-value = 1 × 10^−5^), and CD-HIT (version 4.8.1) was used to filter out redundant genes (C = 0.98) [61].

For phylogenetic analysis of gene families, the protein sequences of the OR family in each species were aligned and trimmed by MAFFT (version 7.487) and TRIMAL (v1.4. rev15) with default options [62,63]. IQ-TREE (version 2.1.4-beta) was used to construct the tree, which was formatted in Evolview [57,64]. Motif analysis was carried out using MEME software (version 5.3.0) with default parameters [65].

### 4.5. Tandem Replication of the ORs in S. invicta

The first transcript of each gene in the genome annotation file was selected as the representative gene sequence and was used as a query in BLAST searches to identify *ORs* derived from tandem duplication and segmental duplication (parameters: blastp, e-value = 1 × 10^−5^). Next, we used MCscanX (version 2.1, Michigan State University, USA) to search for all tandem repeats while using scripts to extract those of *OR* genes [66]. MapChart (version 2.32) was used to construct the chromosomal maps of *M. pharaonis, O. biroi*, and *S. invicta*. A cluster was defined as two or more neighboring *OR* genes separated by a distance of less than 100 kb, and the numbers of clusters were counted.

### 4.6. Genome Synteny Analysis

Whole-genome synteny among *M. pharaonis, O. biroi*, and *S. invicta* was detected and plotted with jcvi, which was a package in Python, with default parameters [67]. The *OR* synteny blocks were colored by appending the specified color code to the *OR* gene in the collinear block file.

### 4.7. Analysis of Evolutionary Selective Pressure

The *OR* protein sequences on each chromosome were aligned by MAFFT, and MEGA [68] was used to construct a neighbor-joining tree. Codon sequence alignments were built using Python scripts. To assess the role of Darwinian positive selection on the evolution of *OR* genes in ants, we used the CodeML program from the PAML (version 4.8a) package to perform positive selection tests using the branch model [56]. The nonsynonymous/synonymous rate ratio (ω = Ka/Ks) was calculated for clusters of 9-exon *OR* genes on each single chromosome in S. invicta. Command settings for the null model were “model = 0, NSsites = 0, fix_omega = 0, omega = 0.8”, and those for the alternative model were “model = 2, NSsites = 0, fix_omega = 1, omega = 0.8”. A Chi-squared test was performed to identify the significant differences between the two models (*p*-value < 0.05).

### 4.8. Analysis of Differential Expression of the ORs in the Antennae of S. invicta Workers from Different Societies

First, we downloaded raw antennae transcriptome data for *S. invicta* workers of monogyne SB/SB and polygyne SB/SB ant colonies from NCBI (GenBank accession number: GSE126684). Next, Trimmomatic (version 0.38, Cambridge University, UK) and Multiqc (version 1.10.1, Cambridge University, UK) were used to review the quality of every sample [69,70]. *S. invicta OR* transcripts were used to construct the *OR* database. Specifically, Salmon with default parameters was used to quantify *OR* transcripts for each sample [71]. Transcripts-per-million (TPM) values of all *ORs* in each sample were extracted from the total quantitative results. Finally, the R package DEGseq2 was used to standardize the data and identify differentially expressed genes (DEGs) using the criteria *p* < 0.05, FDR < 0.001, and |log2FoldChange| > 2. A base-mean TPM value greater than 80 was used as the criterion for the high expression of the 9-exon *ORs*. The results were visualized as a heatmap generated with Evolview (version v2, Northeastern University, Boston, MA, USA) [57].

### 4.9. Expression Analysis Using Semi-Quantitative RT-PCR

Semi-quantitative RT-PCR was performed to verify the expression of *OR* genes. Antennae (500 individual) and legs (300 individual) were collected from monogyne individuals from five castes of the polygyne-type *S. invicta*, namely workers for foragers (WF), reserves (WR), and nurses (WN), alate virgin queen (AVQ), and males (M) [18,72]. Total RNA was extracted as described in [73]. The cDNA was synthesized from total RNA using the Color Reverse Transcription Kit (with gDNA Remover) (EZBioscience, Roseville, California, USA). Gene-specific primers were designed using Primer Premier 5.0 software (Appendix A) and synthesized by Tsingke Biotechnology Co., Ltd. (Guangzhou, China). Green Tap Mix (Vazyme Biotech Co., Ltd., Nanjing, China) was used for PCR under the following conditions: 95 °C for 5 min; 30 cycles of 95 °C for 15 s, 54 °C for 15 s, and 72 °C for 10 s; and extension at 72 °C for 5 min. The 20 μL reaction system consisted of 10 μL Green Tap Mix, 1 μL Primer F, 1 μL Primer R, 2 μL cDNA, and 6 μL ddH_2_O. RT-PCR products were separated on 1% agarose gels, stained by ethidium bromide, and photographed under a UV light in a Gel Doc XR+ Gel Documentation System with Image Lab Software (version 6.1, Bio-Rad, Hercules, CA, USA).

## 5. Conclusions

In this study, 356, 298, and 306 potentially functional *ORs* were identified in the 3 Formicidae species *S. invicta*, *O. biroi*, and *M. pharaonis*, respectively. We observed that the *S. invicta OR* gene family had expanded via tandem replication and that there was a biased localization of 9-exon *ORs* on the chromosomes. This indicated that tandem replication could be the main mechanism of ant *OR* amplification. The phylogenetic analysis of the ORs of the three Formicidae species indicated that there were highly conserved ORs and species-specific 9-exon OR subfamily members in these species. The antennal expression profiles of the *ORs* of the *S. invicta* workers of different social types showed that three *SinvOR* genes had been differentially expressed among the social types and that three 9-exon *ORs* in *S. invicta* were highly expressed in the antennae of the workers. Five *SinvORs* conserved in the Formicidae species had been differentially expressed between castes. When considered altogether, our findings provided new insights into the function and the evolution of *OR* genes in Formicidae ants.

## Figures and Tables

**Figure 1 ijms-24-06624-f001:**
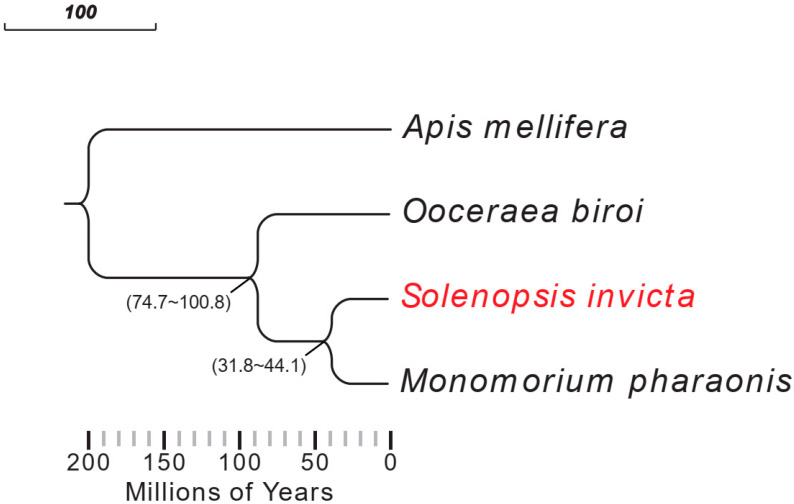
Phylogenetic tree of three Formicidae species. The phylogenetic tree was constructed with Orthofinder2 (version 2.5.4) using the longest transcripts of all genes. *Apis mellifera* were selected as an outgroup. *O. biroi* and *S. invicta* diverged between 74.7 and 100.8 million years ago, and *S. invicta* and *M. pharaonis* diverged between 31.8 and 44.1 million years ago.

**Figure 2 ijms-24-06624-f002:**
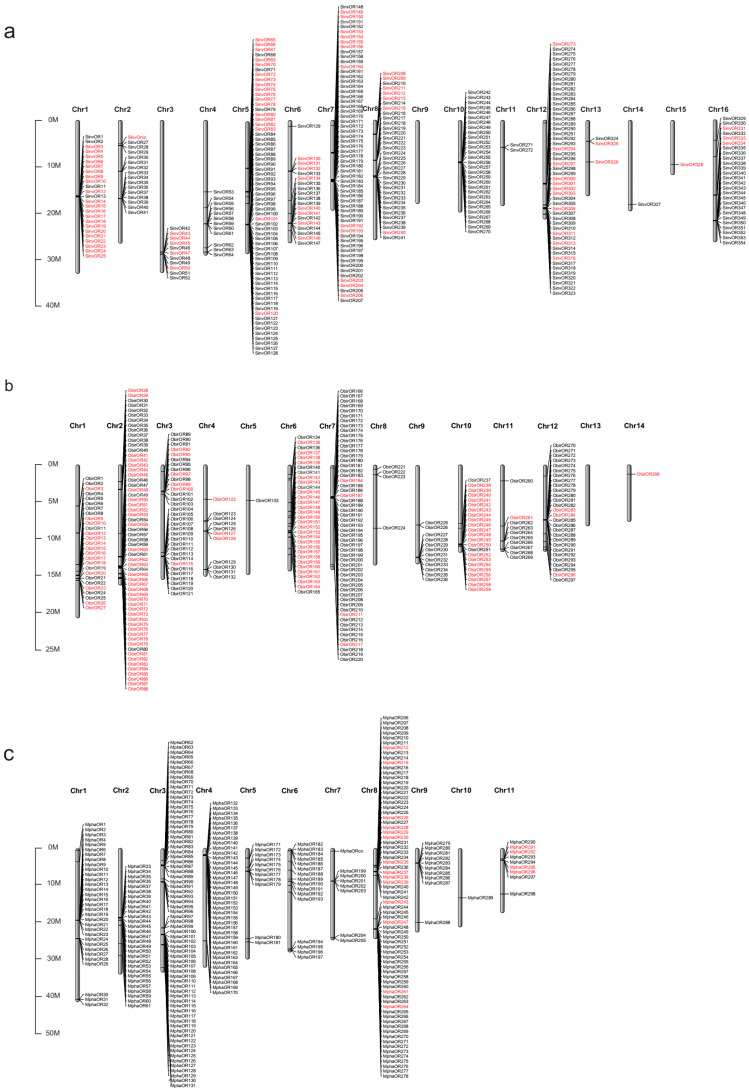
Chromosomal mapping of *OR* genes in three Formicidae species. (**a**–**c**) Chromosomal locations of *OR* genes in *S. invicta* (**a**), *O. biroi* (**b**), and *M. pharaonis* (**c**). The 9-exon *OR* genes are labeled in red.

**Figure 3 ijms-24-06624-f003:**
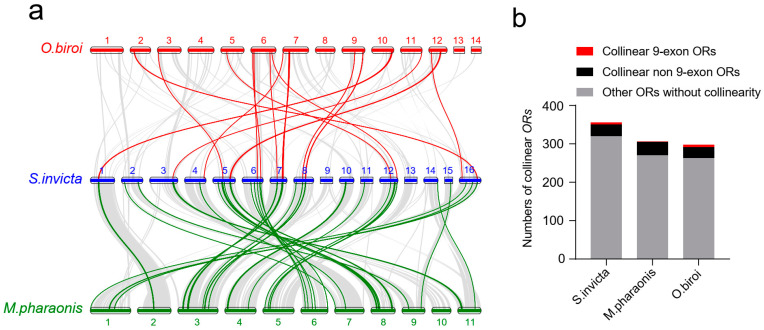
Synteny of *OR* genes in three Formicidae species. (**a**) Genome synteny of *ORs. S. invicta* chromosomes are labeled in blue, *O. biroi* in red, and *M. pharaonis* in green. Syntenic *OR* genes between *S. invicta* and *O. biroi* are connected with a solid red line, and those between *S. invicta* and *M. pharaonis* are connected with a solid green line. Other syntenic genes are connected by a solid gray line. (**b**) Numbers of collinear *ORs* in three Formicidae ant species. The numbers in gray bars represent the *ORs* without genomic collinearity (all other *ORs*), while numbers in red bars (9-exon *ORs*) and black bars (non-9-exon *ORs*) represent all collinear *ORs* in the three ant species.

**Figure 4 ijms-24-06624-f004:**
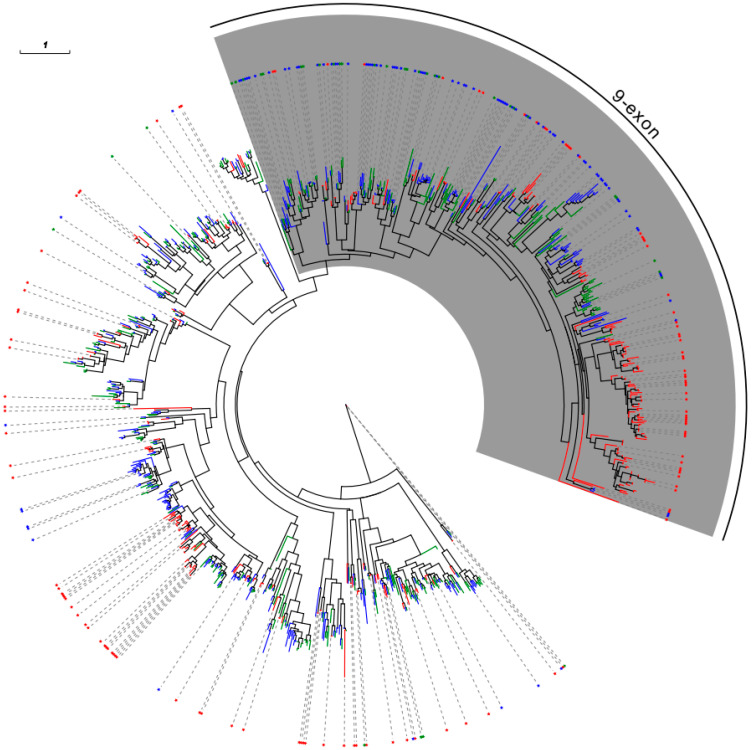
Phylogenetic tree of ORs in three Formicidae species. *ORs* in *S. invicta* are labeled in blue, those in *O. biroi* are labeled in red, in those in *M. pharaonis* are labeled in green. The shaded clade is an expanded clade of the 9-exon *OR* subfamily. The blue star indicates the 9-exon *ORs* of *S. invicta*, the red star indicates the 9-exon *ORs* of *O. biroi*, and the green star indicates the 9-exon *ORs* of *M. pharaonis*.

**Figure 5 ijms-24-06624-f005:**
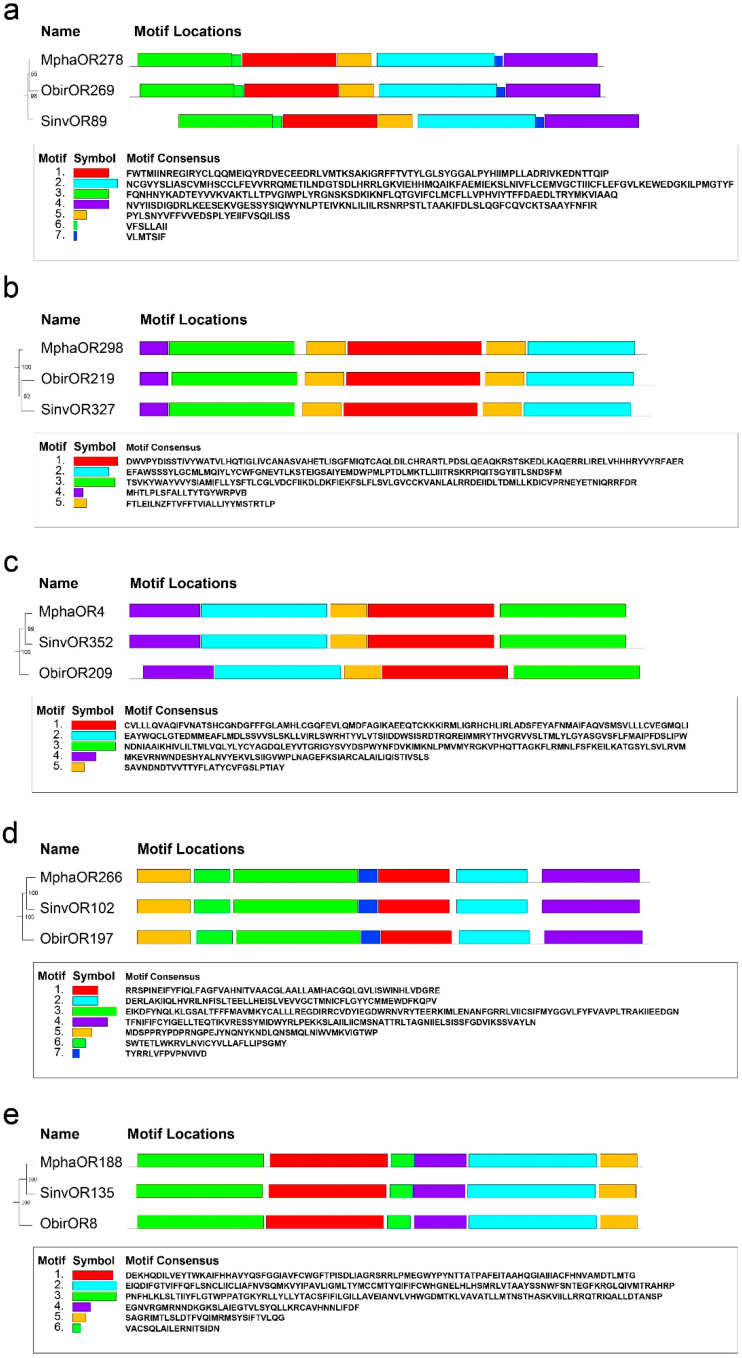
Schematic distribution of motifs found in highly conserved OR proteins in three Formicidae species. They are SinvOR89 (**a**), SinvOR327 (**b**), SinvOR352 (**c**), SinvOR102 (**d**) and SinvOR135 (**e**). Motif analysis was carried out using MEME software (version 5.3.0). The colored boxes represent conserved motifs in each OR.

**Figure 6 ijms-24-06624-f006:**
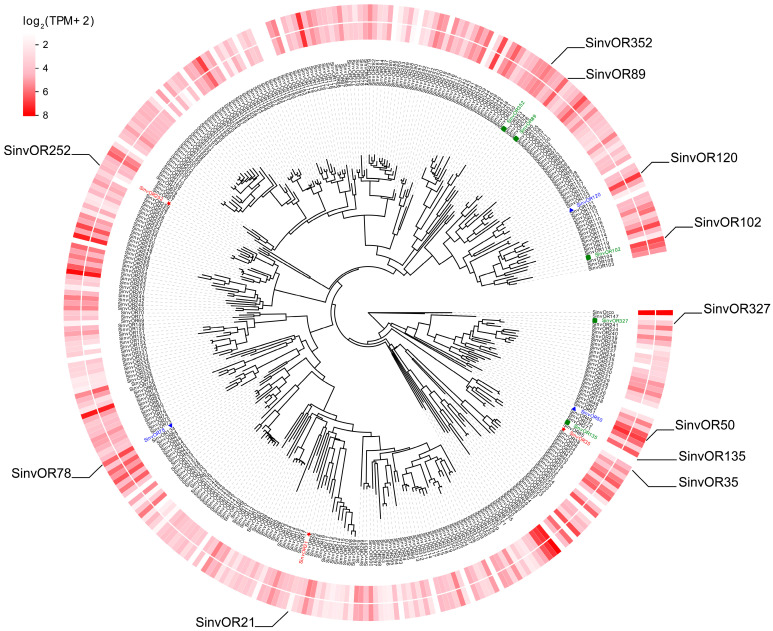
Expression levels of 356 *OR* genes in the antennae of *S. invicta* workers. The inner loop indicates the expression levels of *ORs* in polygyne colonies containing SB/SB workers; outer loop indicates expression in monogyne colonies containing SB/SB workers. Blue font and triangles indicate the highly expressed 9-exon *OR* subfamily members. Red font and pentagons indicate differentially expressed *ORs* in antennae of workers in different social types. Green font and squares indicate highly conserved *ORs* of *S. invicta* in three Formicidae species.

**Figure 7 ijms-24-06624-f007:**
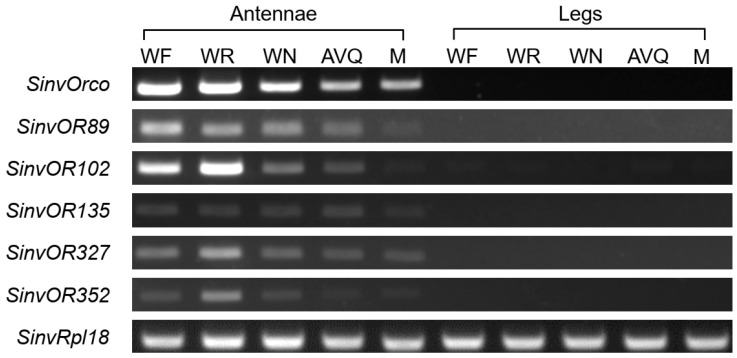
Tissue- and caste-specific expression levels of the conserved *ORs* in *S. invicta*. RT-PCR analysis of *OR* gene expression was performed for antennae (A) and legs (L) of individuals from five castes of *S. invicta*, namely worker for foragers (WF), worker for reserves (WR), worker for nurses (WN), alate virgin queen (AVQ), and males (M). The *Orco* gene was used as the positive control. *Rpl18* was used as a reference gene.

**Table 1 ijms-24-06624-t001:** Selective evolutionary pressure calculated for 9-exon *OR* clades on a single chromosome of *S. invicta*.

Chromosome	Cluster	Ka/Ks	LnL		2Δl (*df* = 1)	*p*-Value
			Null Model	Alternative Model		
Chr1	SinvOR4–8	1.32	−24,468.25	−24,462.34	11.82	5.860 × 10^−4^ **
	SinvOR18–22	0.25	−24,468.25	−24,462.34	11.82	5.860 × 10^−4^ **
Chr3	SinvOR42–47	0.13	−17,005.03	−17,000.66	8.74	3.113 × 10^−3^ *
Chr5	SinvOR65–84	3.10	−82,751.81	−82,751.00	1.62	2.031 × 10^−1^
Chr6	SinvOR130–132	0.17	−14,756.41	−14,742.66	27.5	1.571 × 10^−7^ **
Chr7	SinvOR149–156	0.27	−42,691.96	−42,687.27	9.38	2.194 × 10^−3^ *
Chr8	SinvOR208–215	0.01	−33,648.31	−33,643.12	10.38	1.274 × 10^−3^ *
Chr16	SinvOR331–336	0.10	−25,746.66	−25,745.18	2.96	8.535 × 10^−2^ *

Note: 2Δl: Log likelihood ratios; 2× (lnLH1 − lnLH0). ** significant difference at the level of α = 0.01; * significant difference at the level of α = 0.05.

## Data Availability

The genome of *Apis mellifera* (GCF_003254395.2), *M. pharaonis* (GCF_013373865.1), *O. biroi* (GCF_003672135.1), and *S. invicta* (GCF_016802725.1) were download from NCBI (https://www.ncbi.nlm.nih.gov/, accessed on 14 March 2022). Raw antennae transcriptome data for *S. invicta* workers of monogyne SB/SB and polygyne SB/SB ant colonies was also download from NCBI (GenBank accession number: GSE126684).

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
