# Peer review of "Genome-Wide Analysis of the Odorant Receptor Gene Family in Solenopsis invicta, Ooceraea biroi, and Monomorium pharaonis (Hymenoptera: Formicidae)"

_ijms, 2023, doi:10.3390/ijms24076624_

Round 1

Reviewer 1 Report

Zhang et al., mainly used bioinformatics method to analyze the odorant receptor gene family of three eusocial insects, Solenopsis invicta, Ooceraea biroi, and Monomorium pharaonis based on genomic data. The manuscript found that the S. invicta OR gene family expanded via tandem replication and that there was biased localization of 9-exon ORs on chromosomes. Species-specific 9-exon OR subfamily members were also found. The study seems interesting for the analysis of the relation between eusocial ORs and the CHCs sensory.

Minor revision,

1. What is the relation between ORs tandem repeats and 9-exon ORs?

2. It is not clear that how blue, red, green star shows 9-exon of three eusocial insects in Fig. 4.

3. Fig. 5 showed only 5 or 7 exon, but not 9-exon? Please explain it.

4. Were the red fire ants in the experiment polygyne or monogyne ?

5. Fig. 6 showed 11 SinvORs, but why does the study only use five S. invicta ORs, SinvOR89, SinvOR327, SinvOR135, SinvOR327, and SinvOR135 to be involved in the expressional profile? And why did Fig. 7 use semi quantification PCR to identify the expressional profile rather than real-time PCR?

Author Response

Reviewer 1:

Zhang et al., mainly used bioinformatics method to analyze the odorant receptor gene family of three eusocial insects, Solenopsis invicta, Ooceraea biroi, and Monomorium pharaonis based on genomic data. The manuscript found that the S. invicta OR gene family expanded via tandem replication and that there was biased localization of 9-exon ORs on chromosomes. Species-specific 9-exon OR subfamily members were also found. The study seems interesting for the analysis of the relation between eusocial ORs and the CHCs sensory.

Minor revision,

  1. What is the relation between ORs tandem repeats and 9-exon ORs?

Response: We observed that some of the 9-exon ORs were generated by tandem duplication, particularly on chromosome 1; however, not all 9-exon ORs were replicated in this manner. As shown in the figure below. Therefore, we suggest that although tandem duplication played a significant role in the expansion of the Solenopsis invicta OR gene family, it was not directly correlated with the expansion of 9-exon ORs in S. invicta.

This figure shows the mapping of OR genes on the chromosomes of S. invicta. The locations of the OR genes are indicated on each chromosome, with the 9-exon genes highlighted in red and ORs generated by tandem replication shaded in gray.

  1. It is not clear that how blue, red, green star shows 9-exon of three eusocial insects in Fig. 4.

Response: Figure 4 displays the phylogenetic tree of ORs in three Formicidae species, with stars marking 9-exon ORs. The blue star indicates 9-exon ORs of S. invicta, the red star represents the 9-exon ORs of O. biroi, and the green star indicates the 9-exon ORs of M. pharaonis.

  1. Fig. 5 showed only 5 or 7 exons, but not 9-exon? Please explain it.

Response: We are sorry for the confusion. This figure presents the motif of conserved ORs in three ant species. However, these five conserved OR genes are not 9-exon ORs, and the colored boxes do not represent exons, rather than the conserved domain sequences in each species.

  1. Were the red fire ants in the experiment polygyne or monogyne?

Response: The invasive red fire ants in China are mainly monogynous, with relatively few reports of polygynous ants. In our experiment, monogyne social type of S. invicta was used to do RT-PCR test. We have revised it. Please see line 451.

  1. Fig. 6 showed 11 SinvORs, but why does the study only use five S. invicta ORs, SinvOR89, SinvOR327, SinvOR135, SinvOR327, and SinvOR135 to be involved in the expressional profile? And why did Fig. 7 use semi quantification PCR to identify the expressional profile rather than real-time PCR?

Response: As we mentioned above, the invasive red fire ants in China are mainly monogynous, with relatively few reports of polygynous ants. Strictly speaking, we can hardly find polygyne social type in China, which makes it difficult to validate ORs that display expression differences between monogyne and polygyne colonies at the transcriptional level. Collecting enough tissue for an RT-qPCR analysis was challenging, which is why we did not use this method in our study. Therefore, we selected five conserved ORs to analyze their expression levels, which are in accordance with RNA sequencing.

Reviewer 2 Report

In this study the authors annotated the OR genes from public available genome sequences of 3 ant species and mapped then to chromosomes. They annotated 356 OR in S. invicta (59 of then not previously annotated), 504 alternative spliced OR in O. biroi (Instead of the 503 previously described) and 306 new OR in M. pharaonic. They found that many of them are organized in cluster along chromosomes (especially in the 9-exon OR subfamily) probably caused by tandem duplications, and some of them present synteny in the three species. They also performed a phylogenetic study with the sequences of these ORs, showing that there is an expanded clade of 9-exon ORs from the 3 species, and 17 clades of single-copy orthologs, 5 of then with more than 70% of similarities. Finally, the authors studied differences in antennal expression levels of ORs in workers ants from monogyne SB/SB and polygyne SB/SB colonies of S. invicta using published data. They discovered 3 ORs that present differences of expression between the two types of colonies, and 3 ORs of the 9-exon OR subfamily that are highly expressed in both colony types. And when they studied the 5 ORS from the highly conserved clades, 3 of then are expressed at a relatively high level, while the other 2 have low expression, these five ORs present different antennal expression in 5 castes of ants using RT-PCR.

This study confirms previous findings on the expanding OR gene family in ants and its unique 9-exon OR subfamily using 3 ant species. But the main interest of the article is the new data on the expression of olfactory receptors in the red fire ant (S. invicta), one of the most destructive invasive ant species, which could be of interest in the control of this pest. I also find remarkable the use of publicly available genomic and expression data for the study of specific gene families as the authors have done in this article.  Although the study is well written, properly designed and analyzed and its results and conclusions quite interesting, there are some issues that should be addressed before publication:

- My main concern relates to the RT-PCR experiment depicted in Figure 7, although the results obtained with the 5 conserved ORs in the 5 different clades of S. invicta are very interesting. Why didn't the authors also use in this study the 3 ORs they found to have differential expression in monogynous and polygynous colonies or/and the three 9-exon ORs that are expressed at much higher levels? Furthermore, if they wanted to confirm the result obtained in the expression comparison between monogynous and polygynous workers in Figure 6, they should have compared individuals from both types of colonies in this RT-PCR experiment. In my opinion, the use of the other mentioned OR genes in this experiment and the comparison between both types of colonies could greatly improve this article.  Finally, in the material and method section they do not say the number of individuals of each caste used in this experiment nor whether they came from a monogynous or polygynous colony.

Specific comments:

Page 2, lines 48-49: The authors fail to define paralogs genes as they did with orthologs and Xenologs genes.

Page 2, line 65: Cuticle should be used instead of epidermis.

Page 3, lines 99-107: The authors seem to mix the B/b alleles of the Gp-9 gene with the SB/Sb haplotypes from the non-recombinant genomic region that contains 500 additional genes. They should rephrase the paragraph to make it clear.

Page 8, Figure 4: The Figure 4 resolution should be improved.

Page 12, lines 276-280: This paragraph should be refereed to Figure 6.

Author Response

Reviewer 2:

In this study the authors annotated the OR genes from public available genome sequences of 3 ant species and mapped then to chromosomes. They annotated 356 OR in S. invicta (59 of then not previously annotated), 504 alternative spliced OR in O. biroi (Instead of the 503 previously described) and 306 new OR in M. pharaonic. They found that many of them are organized in cluster along chromosomes (especially in the 9-exon OR subfamily) probably caused by tandem duplications, and some of them present synteny in the three species. They also performed a phylogenetic study with the sequences of these ORs, showing that there is an expanded clade of 9-exon ORs from the 3 species, and 17 clades of single-copy orthologs, 5 of them with more than 70% of similarities. Finally, the authors studied differences in antennal expression levels of ORs in workers ants from monogyne SB/SB and polygyne SB/SB colonies of S. invicta using published data. They discovered 3 ORs that present differences of expression between the two types of colonies, and 3 ORs of the 9-exon OR subfamily that are highly expressed in both colony types. And when they studied the 5 ORS from the highly conserved clades, 3 of then are expressed at a relatively high level, while the other 2 have low expression, these five ORs present different antennal expression in 5 castes of ants using RT-PCR.

This study confirms previous findings on the expanding OR gene family in ants and its unique 9-exon OR subfamily using 3 ant species. But the main interest of the article is the new data on the expression of olfactory receptors in the red fire ant (S. invicta), one of the most destructive invasive ant species, which could be of interest in the control of this pest. I also find remarkable the use of publicly available genomic and expression data for the study of specific gene families as the authors have done in this article.  Although the study is well written, properly designed and analyzed and its results and conclusions quite interesting, there are some issues that should be addressed before publication:

- My main concern relates to the RT-PCR experiment depicted in Figure 7, although the results obtained with the 5 conserved ORs in the 5 different clades of S. invicta are very interesting. Why didn't the authors also use in this study the 3 ORs they found to have differential expression in monogynous and polygynous colonies or/and the three 9-exon ORs that are expressed at much higher levels? Furthermore, if they wanted to confirm the result obtained in the expression comparison between monogynous and polygynous workers in Figure 6, they should have compared individuals from both types of colonies in this RT-PCR experiment. In my opinion, the use of the other mentioned OR genes in this experiment and the comparison between both types of colonies could greatly improve this article.  Finally, in the material and method section they do not say the number of individuals of each caste used in this experiment nor whether they came from a monogynous or polygynous colony.

Response: Thank you for your valuable suggestions. In fact, the invasive red fire ants in China are mainly monogynous, with relatively few reports of polygynous ants. Strictly speaking, we can hardly find polygyne social type in China, which makes it difficult to validate ORs that display expression differences between monogyne and polygyne colonies at the transcriptional level. Thus, we only verified them in monogyne colony. In total, antennae (500 individual) and legs (300 individual) were collected from monogyne social form individuals from five castes of polygyne-type S. invicta. We have added them in line 451.

Specific comments:

Page 2, lines 48-49: The authors fail to define paralogs genes as they did with orthologs and Xenologs genes.

Response: We have provided a more detailed description of the definition of paralogous genes in our article. (lines 47-50).

Page 2, line 65: Cuticle should be used instead of epidermis.

Response: We have revised it. (line 66).

Page 3, lines 99-107: The authors seem to mix the B/b alleles of the Gp-9 gene with the SB/Sb haplotypes from the non-recombinant genomic region that contains 500 additional genes. They should rephrase the paragraph to make it clear.

Response: Thanks for your suggestion. We have revised it in the paragraph to make it clear. (lines 101-104).

Page 8, Figure 4: The Figure 4 resolution should be improved.

Response: Thanks for your suggestion. This image is a high-resolution file with a dpi of 300.

Page 12, lines 276-280: This paragraph should be refereed to Figure 6.

Response: This Figure is relevant to the selective evolutionary pressure on 9-exon OR clusters, rather than the gene expression level.

Reviewer 3 Report

The olfactory systems of eusocial insects, particularly ants, play a crucial role in distinguishing various chemical signals. Odorant receptors (ORs) are vital for detecting these odors, and ants have a significant expansion of OR genes. This study focuses on reannotating OR genes in three ant species, including Solenopsis invicta, to analyze the evolution and function of ORs. The analysis identified potential functional ORs, indicating that tandem duplication was the primary contributor to the expansion of the OR gene family in S. invicta. The study also identified caste-specific expression of ORs in S. invicta workers and showed that some ORs had biased chromosome localization patterns. These findings contribute to our understanding of the OR gene family in ants and the evolution and function of ORs in Formicidae species. Overall, this study contributes to the understanding of the evolution and functions of ORs in ants and provides valuable insights into the olfactory systems of eusocial insects. However, I have the following concerns.

Sample size: The study analyzed three ant species, which is a relatively small sample size for comparative genomic studies. A larger sample size, including more ant species, could increase the power of the analysis and provide a more comprehensive understanding of OR gene evolution and function in ants.

RNA sequencing: The study used RNA sequencing to identify OR genes, which is a commonly used method. However, RNA sequencing can be limited by the quality and quantity of RNA extracted, as well as the completeness of the genome assembly. Using multiple RNA sequencing replicates and improving the genome assembly could increase the accuracy and completeness of the OR gene annotation.

Validation: The study validated the expression of a subset of OR genes using qPCR. While this is a valuable technique, it would be ideal to also validate the expression of OR genes at the protein level, either through immunohistochemistry or other techniques.

While the study provides interesting insights into the differential expression of OR genes in different castes and social types of S. invicta, it is important to note that this is a correlational finding and does not necessarily imply causation. Further research is needed to determine the functional significance of these differential expression patterns.

Author Response

The olfactory systems of eusocial insects, particularly ants, play a crucial role in distinguishing various chemical signals. Odorant receptors (ORs) are vital for detecting these odors, and ants have a significant expansion of OR genes. This study focuses on reannotating OR genes in three ant species, including Solenopsis invicta, to analyze the evolution and function of ORs. The analysis identified potential functional ORs, indicating that tandem duplication was the primary contributor to the expansion of the OR gene family in S. invicta. The study also identified caste-specific expression of ORs in S. invicta workers and showed that some ORs had biased chromosome localization patterns. These findings contribute to our understanding of the OR gene family in ants and the evolution and function of ORs in Formicidae species. Overall, this study contributes to the understanding of the evolution and functions of ORs in ants and provides valuable insights into the olfactory systems of eusocial insects. However, I have the following concerns.

Sample size: The study analyzed three ant species, which is a relatively small sample size for comparative genomic studies. A larger sample size, including more ant species, could increase the power of the analysis and provide a more comprehensive understanding of OR gene evolution and function in ants.

Response: Thank you for your suggestions. Despite the availability of 59 publicly available ant genomes in the NCBI database, only three high-quality genomes are currently assembled at the chromosome level, including Solenopsis invicta, Ooceraea biroi, and Monomorium pharaonis. These ant species are from different subfamilies and have a range of evolutionary distances from the three existing ant genomes, which will provide a more diverse set of data for comparative genomic analysis. In addition, we also used transcriptome data to supplement our comparative genomic analysis. Although the sample number of ant genomes is still relatively small, we believe that our approach of selecting high-quality genomes from different subfamilies provides valuable insights into the evolution and comparative analysis.

RNA sequencing: The study used RNA sequencing to identify OR genes, which is a commonly used method. However, RNA sequencing can be limited by the quality and quantity of RNA extracted, as well as the completeness of the genome assembly. Using multiple RNA sequencing replicates and improving the genome assembly could increase the accuracy and completeness of the OR gene annotation.

Response: Thank you for your suggestions. As you mentioned, obtaining a sufficient sample size is challenging. Thus, in this study, we used the public available data about the raw antennae transcriptome for S. invicta workers downloaded from NCBI (GenBank accession number: GSE126684). We used RNA sequencing data in public with enough replicates to support our comparative genomic analysis.

Validation: The study validated the expression of a subset of OR genes using qPCR. While this is a valuable technique, it would be ideal to also validate the expression of OR genes at the protein level, either through immunohistochemistry or other techniques.

Response: Thank you for your valuable suggestions. Immunohistochemistry is a useful technique that can be considered for studying gene localization.

While the study provides interesting insights into the differential expression of OR genes in different castes and social types of S. invicta, it is important to note that this is a correlational finding and does not necessarily imply causation. Further research is needed to determine the functional significance of these differential expression patterns.

Response: Thank you for your valuable suggestions. In our study, we have identified several candidate OR genes in ants. Next step, we will conduct functional characterization using some technics, such as the heterologous expression system combined with two-electrode voltage clamp, etc.
